# HuR (ELAVL1) regulates the CCHFV minigenome and HAZV replication by associating with viral genomic RNA

**Moe Ikegawa[1,2], Norisuke Kano[2], Daisuke Ori[2], Mizuki Fukuta[3], Minato Hirano[3], Roger Hewson[4], Kentaro Yoshii[3], Taro Kawai[2,5], Takumi Kawasaki[1]***

**1** Immune Dynamics in Viral Infections, National Research Center for the Control and Prevention of Infectious Diseases, Nagasaki University, Nagasaki, Japan, **2** Laboratory of Molecular Immunobiology, Division of Biological Science, Graduate School of Science and Technology, Nagasaki, Japan, **3** Viral Ecology, National Research Center for the Control and Prevention of Infectious Diseases, Nagasaki University, Nagasaki, Japan, **4** London School of Hygiene & Tropical Medicine, Keppel Street, London, UK; and UK-Health Security Agency, Porton Down, Salisbury, United Kingdom, **5** Life Science Collaboration Center (LiSCo), Nara Institute of Science and Technology (NAIST), Ikoma, Japan

* kawast01@nagasaki-u.ac.jp

**Data Availability Statement:** All relevant data are within the paper and its supporting information

**Funding:** This work was supported by Japan Ministry of Education, Culture, Sports, Science and

## Abstract

Crimean-Congo Hemorrhagic Fever virus (CCHFV) is a tick-borne pathogen that causes severe acute fever disease in humans and requires a biosafety level 4 laboratory for handling. Hazara virus (HAZV), belonging to the same virus genus as CCHFV, does not exhibit pathogenesis in humans. To investigate host RNA-binding proteins (RBPs) that regulate CCHFV replication, we generated a series of mutant RAW264.7 cells by CRISPR/Cas9 system and these cells were infected with HAZV. The viral titers in the supernatant of these cells was investigated, and HuR (ELAVL1) was identified. HuR KO RAW264.7 cells reduced HAZV replication. HuR is an RBP that enhances mRNA stability by binding to adenyl-uridine (AU)-rich regions in their 3′ non-coding region (NCR). HuR regulates innate immune response by binding to host mRNAs of signaling molecules. The expression of cytokine genes such as *Ifnb*, *Il6*, and *Tnf* was reduced in HuR KO cells after HAZV infection. Although HuR supports the innate immune response during HAZV infection, we found that innate immune activation by HAZV infection did not affect its replication. We then investigated whether HuR regulates HAZV genome RNA stability. HAZV RNA genome was precipitated with an anti-HuR antibody, and HAZV genome RNA stability was lowered in HuR KO cells. We found that HuR associated with HAZV RNA and stabilized it to enhance HAZV replication. Furthermore, HuR-deficiency reduced CCHFV minigenome replication. CCHFV is a negative-strand RNA virus and positive-strand RNA is produced during replication. HuR was associated with positive-strand RNA rather than negative-strand RNA, and AU-rich region in 3'-NCR of S segment was responsible for immunoprecipitation with anti-HuR antibody and minigenome replication. Additionally, HuR inhibitor treatment reduced CCHFV minigenome replication. Our results indicate that HuR aids replication of the CCHFV minigenome by associating with the AU-rich region in the 3′-NCR.

Technology KAKENHI Grants-in-Aid for Research Activity (B) 20H03468 (T. Kawai), (C)24K10068 (T. Kawasaki), (Early-Career Scientists) 17K15726 (D. O.), (Early-Career Scientists)23K14546 (N.K.), AMED under Grant Number JP23fm0208101 (T. Kawasaki, K.Y.), JP24fm0208101 (T.Kawasaki, K. Y.) and Takeda Science Foundation (T.Kawasaki). N.K. was supported by a Kibou Projects scholarship from the Japanese Society for Immunology. This work was supported by JST, the establishment of university fellowships towards the creation of science technology innovation, Grant Number JPMJFS2137 (M.I.). The funders had no role in study design, data collection and analysis, decision to publish, or preparation of the manuscript.

**Competing interests:** The authors have declared that no competing interests exist.

## Author summary

Crimean-Congo Hemorrhagic Fever virus (CCHFV) is a tick-borne pathogen that causes severe acute fever disease in humans and requires a biosafety level 4 laboratory for handling. Hazara virus (HAZV), the same genus of CCHFV, has been used as a model virus to reveal the molecular mechanism of CCHFV infection. To investigate host factors for CCHFV replication, we generated a series of mutant RAW264.7 cells and conducted screening for host genes that regulate the replication of HAZV, identifying HuR (ELAVL1). HuR is an RNA-binding protein (RBP) that enhances mRNA stability by binding to its 3′ non-coding region (NCR). We found that HuR is associated with the CCHFV RNA stability by binding to its 3'-NCR, and the minigenome assay showed that CCHFV replication is supported by HuR. HuR inhibitor treatment also reduced CCHFV minigenome replication. These findings present a possible starting point for the future development of antiviral drug targeting host RBPs. Combination treatment with RdRp and RBP inhibitors may be a potential therapeutic strategy in the future.

## Introduction

Crimean-Congo hemorrhagic fever virus (CCHFV) belongs to the genus Orthonairovirus and is a tri-segmented negative-sense RNA virus [1–3]. CCHFV is a tick-borne pathogen responsible for severe acute fever with a case fatality rate of 5–30%, and is distributed in Africa, Middle East, southern Europe and Asia [4]. These areas overlap with the distribution of *Hyalamma* spp., which are major tick vectors [4]. CCHFV transmission occurs through tick bites in humans and handling of infected animal blood or tissues. Additionally, the nosocomial route of CCHFV transmission has been reported in several countries [5,6]. Molecular pathogenesis of CCHF remains largely unknown. Several studies have shown that patients with the severe form of CCHF exhibit a high level of inflammatory cytokines, such as TNF, IL-6 and IFN-γ, and the excess of these cytokines may trigger vascular dysfunction, disseminated intravascular coagulation, organ failure and shock [7,8].

CCHFV requires a biosafety level 4 (BSL-4) laboratory for handling and has been declared by the World Health Organization (WHO) as an R&D Blueprint priority pathogen [9]; therefore, research on viral-host interactions and pathogenicity of the virus has been limited. Hazara virus (HAZV) and CCHFV belong to the same genus, and share many structural and biological characteristics. HAZV has been used as a model virus to reveal the molecular mechanism of CCHFV [10–12]. Wild type (WT) mice infected with HAZV did not show pathogenesis, however IFN-α/βR knockout (KO) mice infected with HAZV showed mortality within one week. Histopathological findings in HAZV infected IFN-α/βR KO mice were identified in the liver, spleen and lymph nodes, with changes similar to a mouse model of CCHFV infection [10]. HAZV infection in human cells caused a stronger induction of cytokine gene expression such as *IL6*, *IFNb* and *TNFa* [13]. CCHFV and HAZV have three negative-strand RNA separated into small (S), medium (M), and large (L) segments. S segment encodes nucleoprotein (N), and M segment encodes the precursor that originates the mature glycoproteins of the viral envelope (Gn and Gc) and some additional proteins. L segment encodes L protein containing RNA-dependent RNA polymerase (RdRp). Coding regions of these proteins are located between 5′ and 3′ non-coding regions (NCRs) that are essential as promoter sequences for viral genome replication and synthesis of complementary strand [14–16].

Patients with CCHF exhibit high levels of inflammatory cytokines [7,8] and CCHFV infection to monocyte derived dendritic cells also induces cytokine production [17]. Innate immune responses to viral infection are initiated upon sensing of viral nucleic acids by host pattern-recognition receptors. Negative-strand RNA viruses are sensed by the cytosolic proteins, retinoic acid–inducible gene I (RIG-I) and melanoma differentiation-associated protein 5 (MDA5), which transmit the signal through the mitochondrial protein interferon (IFN)-β promoter stimulator 1 (IPS-1) (also called MAVS), which culminates in the activation of the transcription factors NF-κB and IRF3 [18,19]. After viral infection, IRF3 is phosphorylated by the kinase TBK1 and/or its related kinase IKKi (also known as IKKε) and subsequently translocates into the nucleus [20]. IKKα/β phosphorylate IκB proteins and triggers their degradation, allowing NF-κB to translocate to the nucleus. IRF3 and NF-κB cooperatively regulate the expression of genes encoding pro-inflammatory cytokines and type I IFNs [21,22].

RNA-binding proteins (RBPs) play a crucial role in the post-transcriptional regulation of immune response, cancer, development and other biological events [23,24]. A major strategy for RBPs is the regulation of mRNA stability. Human antigen R (HuR), also known as ELAVL1, is an RBP that increases the stability of host target mRNAs by binding adenyl-uridine (AU)-rich regions in their 3'-NCR [21,25,26]. HuR has three RNA-recognition motifs, and belongs to the Hu protein family, which is composed of HuR, HuB, HuC, and HuD. HuR is ubiquitously expressed, whereas HuB, HuC, and HuD are specifically expressed in the neuronal tissues [27]. Previous reports have shown that HuR maintains the mRNA of various target genes, including IFNB1, COX2, IL8, and TGFB1, which contain the AU-rich and U-rich element in 3'- NCR [28–30]. We have also reported that *Plk2* and *Atp6v0d2* are HuR-target mRNAs that regulate antiviral and inflammatory response [31,32]. HuR-crosslinking and immunoprecipitation with RNA sequences revealed ~26,000 HuR-binding sites [33] and ~3,000 target mRNAs stability [34].

Mouse models of CCHFV infection have provided evidence that hepatocytes and endothelial cells are targets of CCHFV infection, which causes liver damage and vascular dysfunction [3]. Monocytes and macrophages are also infected by CCHFV, which may contribute to disease progression and excessive inflammatory response [35]. In this study, we generated a series of mutant RAW264.7 cells, a macrophage-like cell line used for the immune response [31,32], and conducted screening for host genes that regulate the replication of HAZV, identifying HuR. HuR regulated HAZV replication by association with its RNA genome. Furthermore, the minigenome assay for CCHFV supported that HuR regulates the replication of CCHFV. It is widely accepted that HuR binds to the AU-rich region in the 3′-NCR of host mRNA. Our results indicated that HuR helps the replication of CCHFV minigenome by association with AU-rich region in the 3′-NCR of RNA.

## Materials and methods

### Ethics statement

The experiments were approved by Nagasaki University Recombinant DNA Experiment Safety Committee (approval number 2303291851–5).

### Cells and reagents

HEK293 cells, RAW264.7 cells and MEFs were cultured in DMEM (Nacalai Tesque) supplemented with 10% heat-inactivated FBS in a 5% $CO_2$ incubator. High molecular weight (HMW) polyinosinic-polycytidylic acid [poly(I:C)] was purchased from InvivoGen. Poly(I:C) was mixed with Lipofectamine 2000 (Life Technologies) at a ratio of 1:1 (μg/μL) in Opti-MEM (Life Technologies) and cells were stimulated with 2 μg/mL poly(I:C) for intracellular

stimulation. The transcriptional inhibitor actinomycin D was purchased from Sigma-Aldrich. *Rig-I/Mda5* KO MEFs [18], *IPS1* KO MEFs [19] and *Tbk1/Ikk-i* KO MEFs [20] were generated from each KO mice and were kindly provided by Dr Shizuo Akira (Osaka Univ.). LDH in the culture supernatant was measured using LDH assay kit (nacalai tesque) according to the manufacturer's instructions.

### Generation of mutant cell line

CRISPR-associated protein 9 (Cas9) were cloned into the retroviral expression plasmid LZRS-IresGFP (#21961, Addgene, Waltham, MA, USA) digested with EcoRI and BamHI. The recombinant retroviruses were packaged in Platinum E cells, and used to infect and select RAW264.7 cells. Stable Cas9-expressing RAW264.7 cells were selected by flow cytometry using FACSAria Fusion (BD Bioscience). Guide RNAs (gRNAs) were selected using CRISPR-direct (https://crispr.dbcls.jp/), which were cloned into the lentiviral gRNA expression plasmids pKLV-U6gRNA (BbsI)-PGKpuro2ABFP (#50946, Addgene) digested with BbsI or pgRNA-humanized (#44248, Addgene) digested with BglII. The gRNA-containing lentiviruses were packaged in HEK293T cells and used to infect and select RAW264.7 cells. Stable gRNA-expressing RAW264.7 cells were selected using 4 μg/mL puromycin for 3 days, and then used for the experiment. Generated cloned cells and gene names were listed in S1 Table.

### Virus and viral titration

The HAZV JC280 strain is described in a previous paper [36]. Working stocks of the virus were prepared using SW-13 cells. For titration of HAZV, serial dilutions of the virus were mounted on monolayer SW-13 cells in 24-well plate for 1 h, and SW-13 cells were overlaid with MEM containing 2% (v/v) FBS and 0.7% (w/v) UltraPure Agarose (Thermo Fisher Scientific). After incubation for 3 days, the cells were fixed with ethanol containing 16% (v/v) acetic acid and stained with 1% (w/v) amido black in phosphate–buffered saline (PBS) after removal of overlaid agarose. Tick-borne encephalitis virus (TBEV) strain Oshima 5–10 [37] and Japanese encephalitis virus (JEV) strain JaOArS982 [38] are described in previous papers. Serial dilution of TBEV and JEV was added on the monolayer BHK cells in 24-well plate for 1 hour and cells were overlaid with MEM containing 2% (v/v) FBS and 1.5% (w/v) Carboxymethyl Cellulose (CMC) (Wako). CMC contained medium was removed after incubation for 3 days, and the cells were fixed and stained with 10% formalin with 0.25% crystal violet. HAZV was exposed 30 W UV light (CRF/UV-30A) for 30 minutes (min) for inactivation.

### RNA isolation

Total RNA was extracted with TRI Reagent (Molecular Research Center, Inc.), according to the manufacturer's protocol.

### Real-time PCR assay

Total RNA was reverse transcribed to cDNA using random primers with ReverTra Ace (Toyobo), according to the manufacturer's protocol. KAPA SYBR Green PCR Master Mix (Kapa Biosystems) was used for real-time PCR and the measurements were performed using QuantStudio 3 (Applied Biosystems). The PCR conditions were as follows: 95˚C for 3 min; 40 cycles of 95˚C for 5 seconds (s), 60˚C for 30 s; and melt curve stage. Amplified products were detected at 60˚C. HAZV S (354–697 bp), M (740–1012 bp), and L (4940–5330 bp) were amplified and subcloned as standard DNA. Copy numbers of S, M, and L segments were calculated using a standard curve from the titration of standard DNA. Copy number of CCHFV reporter

RNA was calculated using a standard curve from the reverse transcription of titrated CCHFV reporter RNA. The primer sequences were listed in S2 Table.

### ELISA

RAW264.7 cells were seeded in 96-well plates and infected with 1 MOI HAZV for the indicated time period. The cytokine level of IL-6 in the culture supernatant was measured using Mouse IL6DuoSet ELISA (R&D Systems) according to the manufacturer's instructions.

### Western blotting

Whole-cell lysates were prepared by lysing the cells in 50 mM Tris-HCl (pH 8), 150 mM NaCl, 10 mM EDTA, 2 mM EGTA, 0.25% triton X-100. After sonication and centrifugation at $12,000 \times g$ for 10 min at 4°C, supernatants were collected and used as whole-cell lysates. Whole-cell lysates were subjected to SDS-PAGE and transferred to a polyvinylidene fluoride (PVDF) Immobilon Transfer Membrane (Millipore). The membranes were immunoblotted with the indicated antibodies. Bound antibodies were visualized with horseradish peroxidase (HRP)-conjugated antibodies against mouse or rabbit IgG (Sigma-Aldrich) using Immobilon Forte (Millipore). HRP activity was detected using a LAS 3000 system (Fuji-film). The following primary antibodies were used for western blotting. Rabbit anti-pIRF3 (4947S, Cell Signaling Technology), rabbit anti-pp65 (3033S, Cell Signaling Technology), mouse anti-IRF3 (sc-33641, Santa Cruz Biotechnology), mouse anti-NF-kB p65 (sc-8008, Santa Cruz Biotechnology), mouse anti-HuR (sc-5261, Santa Cruz Biotechnology), and mouse anti-Actinβ (sc-47778, Santa Cruz Biotechnology) antibodies were used. Anti-HAZV N antibody was prepared by immunizing BALB/c mice with recombinant HAZV N protein expressed in *E. coli*. Rabbit anti-pIRF3, rabbit anti-pp65 and mouse anti-HAZV N were diluted 1:1000, and other antibodies were diluted 1:100. Anti-rabbit IgG (A0545-1ML, Sigma-Aldrich) and anti-mouse IgG (A4416-1ML, Sigma-Aldrich) peroxidase antibodies were used at 1:10000 dilution.

### Promoter reporter assay

HEK293 cells were plated in 24-well plates and transiently transfected with 50 ng of IFN-β reporter plasmid or ISRE reporter plasmid and pRL-SV40 (Promega) as an internal control. After 24 h of transfection, cells were stimulated with 1 MOI HAZV or poly(I:C), and luciferase activities were measured after 24 h stimulation with SpectraMaxiD5 (Molecular Devices) using the Dual-Glo Luciferase Assay System (Promega), according to the manufacturer's instructions.

### Minigenome assay

Plasmid construction for the minigenome assay was performed as previously described [15]. Briefly, reporter RNA were synthesized from pUC-GW-Amp-secNluc-CCHFV Lseg or pUC-GW-Amp-secNluc-HAZV Sseg under the T7 promoter *in vitro* (Takara ITVpro mRNA Synthesis Kit). HEK293 cells in 24-well plate were transfected with pCAG-Hyg-CCHFV L, pCAG-Hyg-CCHFV N, and the CCHFV reporter RNA for the CCHFV minigenome assay, using Lipofectamine 2000. Minigenome assay of HAZV was conducted in the same manner. The luciferase activity of secNluc in the supernatant was measured with SpectraMaxiD5 (Molecular Devices) using the Nano-Glo Luciferase Assay System (Promega), according to the manufacturer's instructions.

### RNA stability assay

WT and HuR KO RAW264.7 cells at 48 h post-infection with HAZV were treated with 2.5 μg/mL actinomycin D and incubation was stopped by adding TRI Reagent.

### RNA immunoprecipitation

RAW264.7 cells infected with 0.1 MOI HAZV for 2 days or HEK293 cells transfected with positive-strand or negative-strand CCHFV reporter RNA for 3 h were lysed by pumping using a 27 G syringe needle (Terumo) in polysome lysis buffer (100 mM KCl, 5 mM MgCl$_2$, 10 mM HEPES [pH7], 0.5% Nonident P-40, 1mM DTT). After centrifugation at $12,000 \times g$ for 15 min at 4˚C, the supernatants were incubated in the presence of 2 μg anti-HuR antibody (Santa Cruz Biotechnology) or control IgG (Bio X cells) with Protein-G Sepharose beads (Sigma-Aldrich) for 3 h at room temperature. Beads were washed three times with polysome lysis buffer and TRI Reagent (Molecular Research Center, Inc.) was added for RNA isolation.

### Statistical analysis

Data are expressed as the mean standard deviation (SD). Statistical analyses were performed with R (4.0.3).

## Results

Monocytes and macrophages infected with CCHFV may contribute to disease progression and excessive inflammatory response. To investigate CCHFV replication in macrophages, RAW264.7 cells, a macrophage-like cell line were infected with 0.1 MOI HAZV, and the virus titers in the supernatant were measured by the plaque assay. Mouse embryonic fibroblasts (MEFs) were also infected with HAZV to compare macrophages with non-immune cells. The virus titers of RAW264.7 cells peaked at 24–48 h and sustained to 72 h (Fig 1A). The virus titers of MEFs peaked at 48 h and decreased at 72 h (Fig 1A). To identify host cell factors responsible for CCHFV replication, we generated RAW264.7 cells with a frame-shifted mutation in the exon of genes encoding RBPs. We generated 66 cell lines targeting 35 genes, and for most of the target genes, we generated two mutant cell lines per gene (S1 Table), and virus titers of these cells at 48 h after 0.1 MOI HAZV infection were measured by the plaque assay. The virus titers of these cells were plotted, and the cell lines that showed higher or lower titer than the WT RAW264.7 cells were highlighted (Fig 1B). The virus titers of mutant cells for *Zfp36l1*, *Parp12*, and *Unk* were increased whereas those of mutant cells for Trm1 and HuR were reduced. HuR increases the stability of host target mRNAs by binding to AU-rich regions in their 3'-NCRs [21,25,26] and we have found that HuR-target host mRNAs to regulate innate immune response [31,32]. We selected HuR to investigate whether it also regulates the stability of exogenous RNA during viral replication. HuR deficiency in these mutant cells was confirmed by western blotting with anti-HuR antibody (Fig 1C). WT and HuR KO cells were plated and the number of these cells was counted (S1 Fig). The proliferation of WT and HuR KO cells was not significantly altered. To investigate whether lowering the virus titer in HuR KO cells is specific to HAZV infection, HAZV, TBEV and JEV infected WT RAW264.7 cells and HuR KO cells, and the virus titers were measured at the different time points (Fig 1D). HuR-deficiency lowered the HAZV titers from 48 to 120 h after infection. The TBEV titer in WT and HuR KO cells was not significantly altered, except at 72 h post infection. JEV titer in HuR KO cells was higher than those in WT cells at 48 and 72 h post infection. It is unclear why TBEV titer at 72 h and JEV virus titer at 48 and 72 h increased in HuR KO cells; however, HuR deficiency differentially affected viral replication. To investigate copy number of HAZV inside

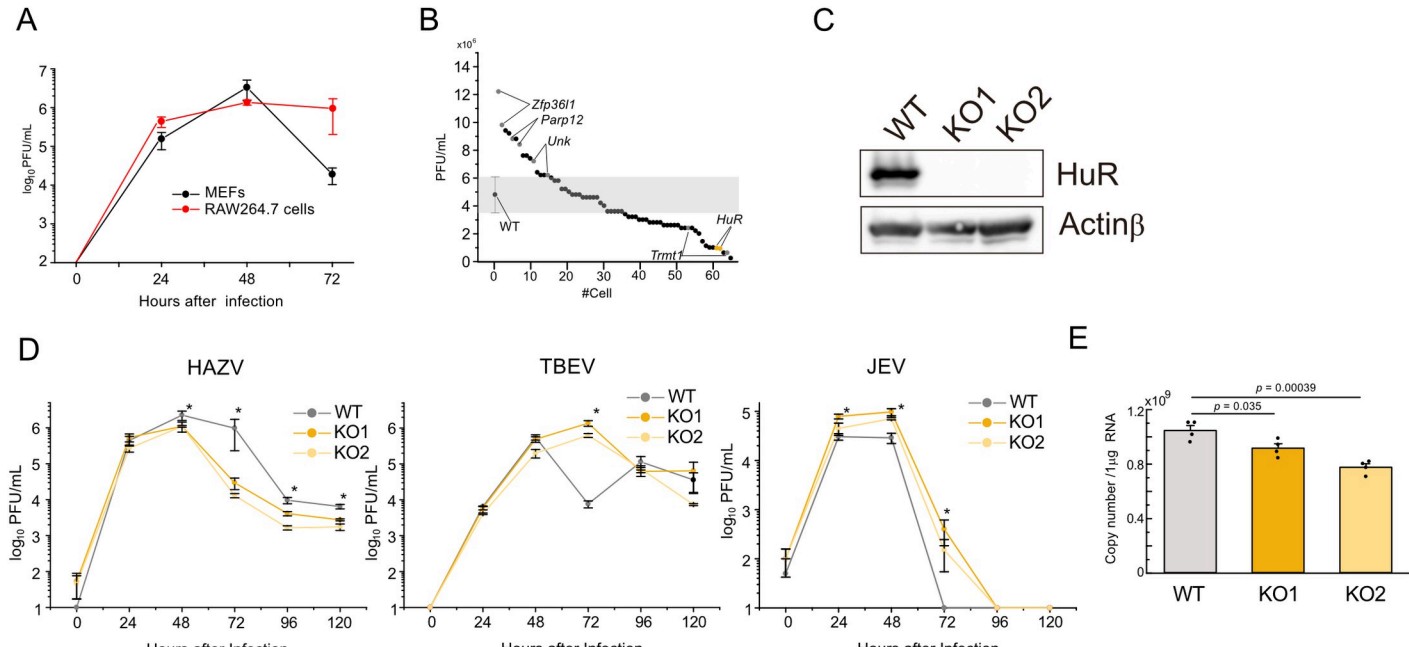

**Fig 1. Identification of HuR as a regulator for HAZV replication in RAW264.7 cells.** A, RAW264.7 cells and MEFs were infected with 0.1 MOI HAZV and virus titers in the supernatant were measured. B, WT and mutant RAW264.7 cells were infected with 0.1 MOI HAZV and the virus titers in the supernatants were measured at 48 h post-infection. Virus titers of the mutant cells were plotted, and cell lines that showed higher or lower titers than WT were highlighted. C, Cell lysates were extracted and immunoblotted using anti-HuR and anti-Actinβ antibodies. D, WT and HuR KO RAW264.7 cells were infected with 0.1 MOI of HAZV, TBEV or JEV, and the time courses of virus titer in the supernatants were measured at the indicated time points. E, The copy number of HAZV S segment inside cells was measured by real-time PCR at 48 h after 0.1 MOI HAZV infection. One-way ANOVA with Tukey's multiple comparison test (C,D); *$p < 0.05$.

cells, RNA was isolated from WT, KO1 and KO2 cells at 48 h after infection with 0.1 MOI HAZV, and the copy number of S segment (N) RNA in HuR KO cells was lower than that in WT cells (Fig 1E).

HuR regulates innate immune response via RIG-I/MDA5-dependent and nucleic acids-sensing endosomal Toll-like receptors-dependent pathways [31,32]. CCHFV and HAZV infection caused activation innate immune response to induce expression of cytokine genes [7,8,17]. We investigated innate immune regulation by HuR during HAZV and JEV infection. WT and HuR KO RAW264.7 cells were infected with 1 MOI HAZV and at 9 h post infection, the expression of *Il6*, *Ifnb*, *Tnf*, *Cxcl10*, *Il10*, *Il12p40*, *Ccl2* and *Tgfb* relative to non-infected control cells and the copy number of S segment inside cells were measured (Fig 2A and 2B). HAZV was exposed to UV for inactivation, and WT and HuR KO RAW264.7 cells were also stimulated with UV-treated HAZV. The expression of *Il6*, *Ifnb*, *Tnf*, and *Cxcl10* after HAZV infection was significantly increased in WT cells, and was not altered by UV-treated HAZV (Fig 2A). *Ifnb*, *Il6*, *Tnf* and *Cxcl10* expression was reduced in HuR KO cells compared to WT cells. The copy number of S segment inside cells increased after 1 MOI HAZV infection, but not after UV-treated HAZV treatment (Fig 2B). To compare cytokine expression with JEV infection, WT and HuR KO RAW264.7 cells were infected with 10 MOI JEV and the expression of these cytokine genes was measured (S2 Fig). *Tnf* and *Ccl2* expression was only slightly increased in WT cells after JEV infection, suggesting that viral proteins in JEV reduced innate immune response. IL-6 production in the supernatant did not increase at 9 h post infection, whereas it increased at 24 h post infection and was lower in HuR KO cells than in WT cells (Fig 2C). Then, the phosphorylation of transcription factors IRF3 and p65 (RelA), a major component of NF-κB, was measured after the infection. Phosphorylation of IRF3 level was

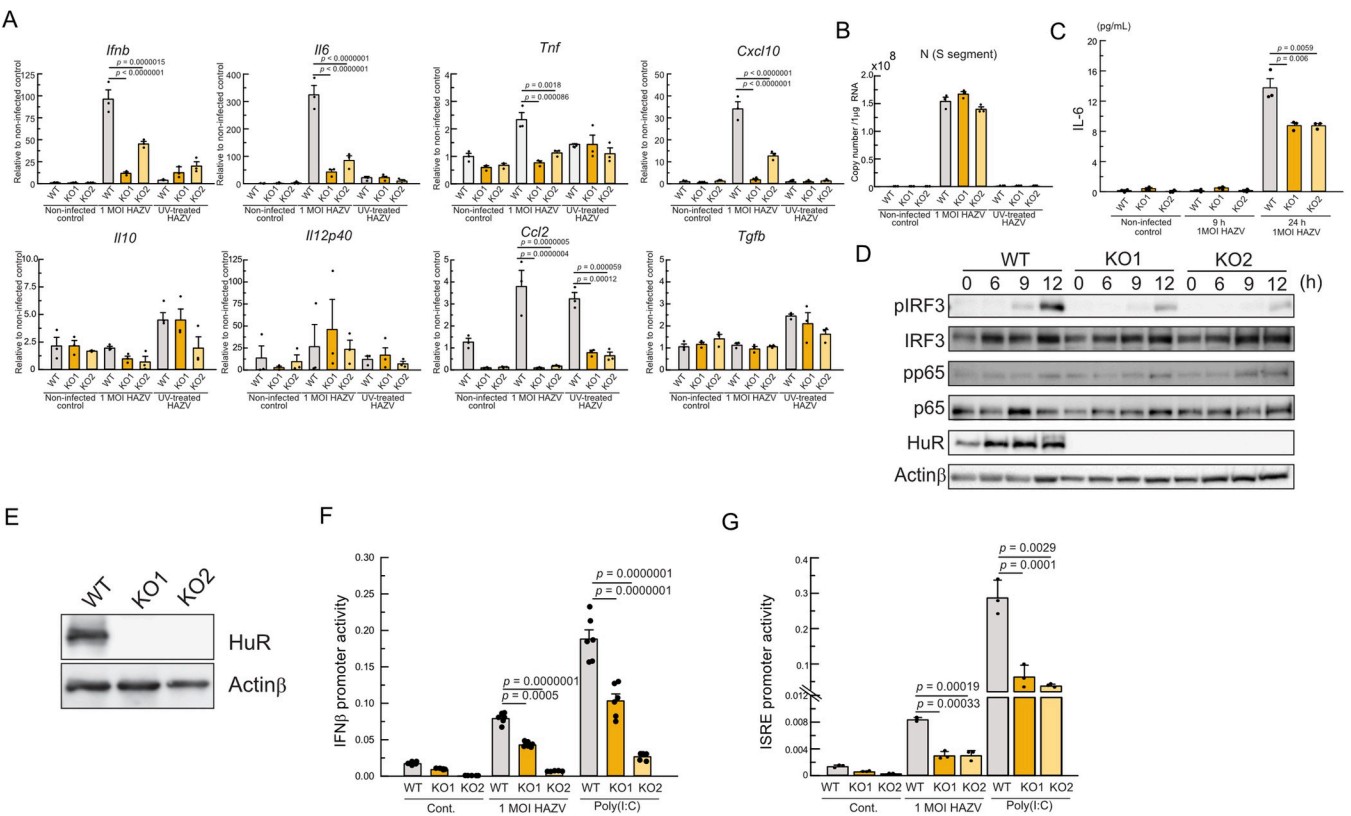

**Fig 2. Innate immune response after HAZV infection in HuR KO cells.** A,B, WT and HuR KO RAW264.7 cells were infected with 1 MOI HAZV or UV-treated HAZV, and cytokine gene expression was measured at 9 h after infection by real-time PCR (A). Gene expression was calculated by fold increase compared to non-infected control WT cells. Copy number of S segment inside cells was calculated by real-time PCR (B). C, IL-6 production in the supernatant was measured by ELISA after 1 MOI HAZV infection. D, WT and HuR KO RAW264.7 cells were infected with 1 MOI HAZV. Cell lysates were extracted and immunoblotted with the indicated antibodies. E, HuR KO HEK293 cells were generated by genome editing and the deficiency was confirmed by western blotting. F,G, WT and HuR KO HEK293 cells were transfected with a reporter plasmid driven by IFN-β promoter (F) or ISRE promoter (G) with internal control promoter plasmid, and these cells were stimulated with 1 MOI HAZV or poly(I:C). Luciferase activity was measured 24 h after stimulation. The ratio of intensity of firefly luciferase (IFN-β/ISRE promoter) to intensity of renilla luciferase (internal control promoter) was plotted. One-way ANOVA with Tukey's multiple comparison test (A,C,F,G).

increased from 9 h after infection and was lowered in HuR KO cells, whereas phosphorylation of p65 was slightly increased in both WT and HuR KO cells (Fig 2D). To perform luciferase promoter assay, we generated HuR-deficient HEK293 cells, and the deficiency was confirmed by western blot analysis using an anti-HuR antibody (Fig 2E). The IFN-β promoter and ISRE promoter activity at 24 h after 1 MOI HAZV infection or poly(I:C) stimulation were reduced in HuR KO cells (Fig 2F and 2G). These results indicated that the innate immune response to HAZV infection was reduced by HuR-deficiency.

HAZV infection triggers activation of innate immune response in mouse macrophages (Fig 2) and genomic RNA in negative-stranded RNA viruses has been reported to be recognized by RIG-I/MDA5 [18,19]. HuR regulates RIG-I/MDA5-dependent innate immune response [31]. Innate immune suppression by HuR deficiency expects the increase of viral replication (Fig 2), however the HAZV titer was lowered in HuR KO cells (Fig 1). To further investigate the role of innate immune activation by HAZV infection in viral replication, we tested whether innate immunity was activated by HAZV through the RIG-I/MDA5 signaling pathway. WT, *Rig-I/Mda5* KO, *IPS1* KO, and *Tbk1/Ikk-i* KO MEFs were infected with 1 MOI HAZV, and *Ifnb*, *Il6* and *Tnf* expression was measured at 24 h after infection (Fig 3A). Expression of these genes

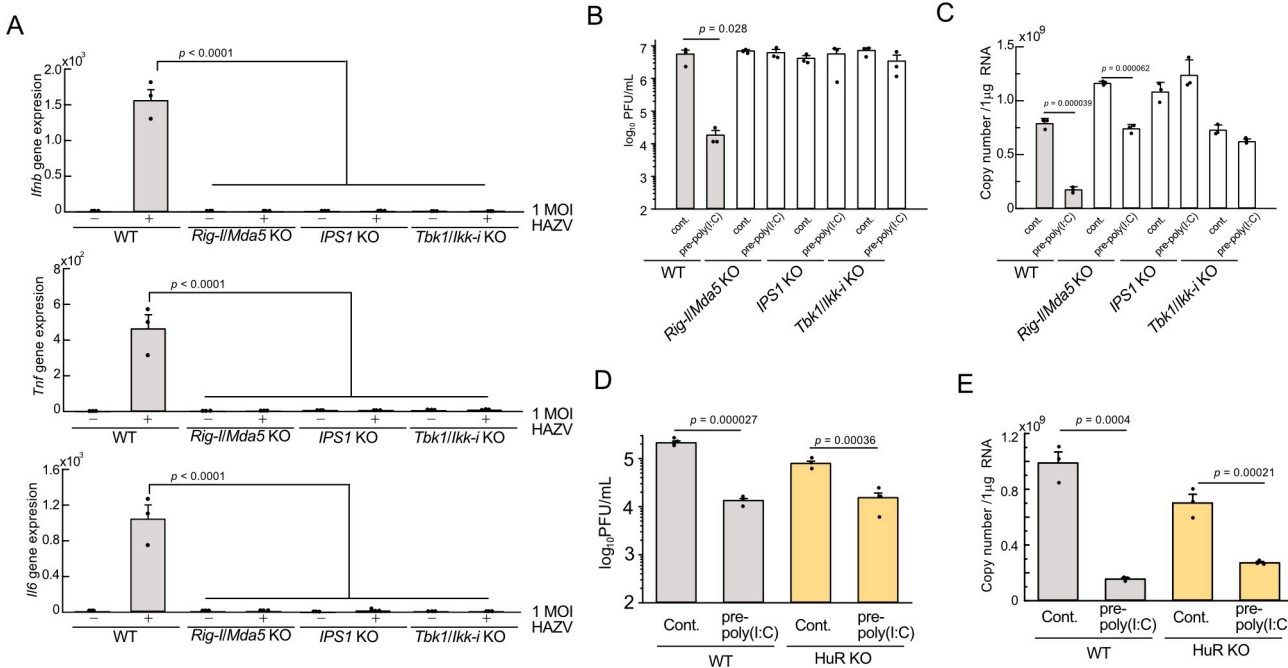

**Fig 3. Innate immune activation by HAZV infection is not involved in its replication.** A, WT, *Rig-I/Mda5* KO, *IPS1* KO and *Tbk1/Ikk-i* KO MEFs were infected with 1 MOI HAZV and the expression of cytokine genes was measured by real-time PCR at 24 h post infection. B,C, WT and KO MEFs with or without pre-poly(I:C) for 6 h were infected with 0.1 MOI HAZV, and virus titers in the supernatant (B) or copy number of S segment inside cells (C) were measured at 48 h post-infection. D,E, WT and HuR KO RAW264.7 cells with or without pre-poly(I:C) for 6 h were infected with 0.1 MOI HAZV, and virus titers in the supernatant (D) or copy number of S segment inside cells (E) were measured at 48 h post-infection. Ono-way ANOVA with Tukey's multiple comparison test (A, C).

was significantly increased in WT infected cells contrary to KO cells, indicating that HAZV triggers innate immune response in RIG-I/MDA5-dependent manner. Poly(I:C) transfection robustly activates RIG-I/MDA5-dependent signaling. To further investigate the role of the innate immune response in viral production, WT and those KO MEFs were infected with 0.1 MOI HAZV with or without poly(I:C) pre-stimulation for 6 h (pre-poly(I:C)). The virus titers of HAZV in the supernatant of WT increased to around $1 \times 10^6$ PFU/mL at 48 h post infection and were comparable among those KO MEFs (Fig 3B), indicating the innate immune response induced by HAZV doesn't affect virus replication. In contrast, the virus titer in WT MEFs was lowered by pre-poly(I:C) treatment, and the virus titer of those KO MEFs was not altered by pre-poly(I:C) treatment. The copy number of HAZV inside cells also supported these results (Fig 3C). To further investigate the role of the innate immune response in RAW264.7 cells, RAW264.7 WT and HuR KO cells were pre-stimulated with poly(I:C) for 6 h [31] and infected with HAZV (Fig 3D and 3E). Pre-poly(I:C) treatment in both WT and HuR KO cells reduced the virus titer (Fig 3D) and the copy number of viral RNA inside cells (Fig 3E) compared to the control. These findings indicated that activation of innate immunity by poly(I:C) lowered HAZV replication, however innate immune activation by HAZV infection doesn't affect its replication.

HuR is known as an RBP that stabilizes the target mRNAs, and we hypothesized that HuR is involved in the stabilization of HAZV RNA. To test whether HAZV genome replication is diminished by HuR-deficiency, the copy number of HAZV RNA for the S, M and L segments inside of WT and HuR KO cells was measured by real-time PCR (Fig 4A). The copy number of all three segments at 24 h post infection was reduced in HuR KO cells and the copy number

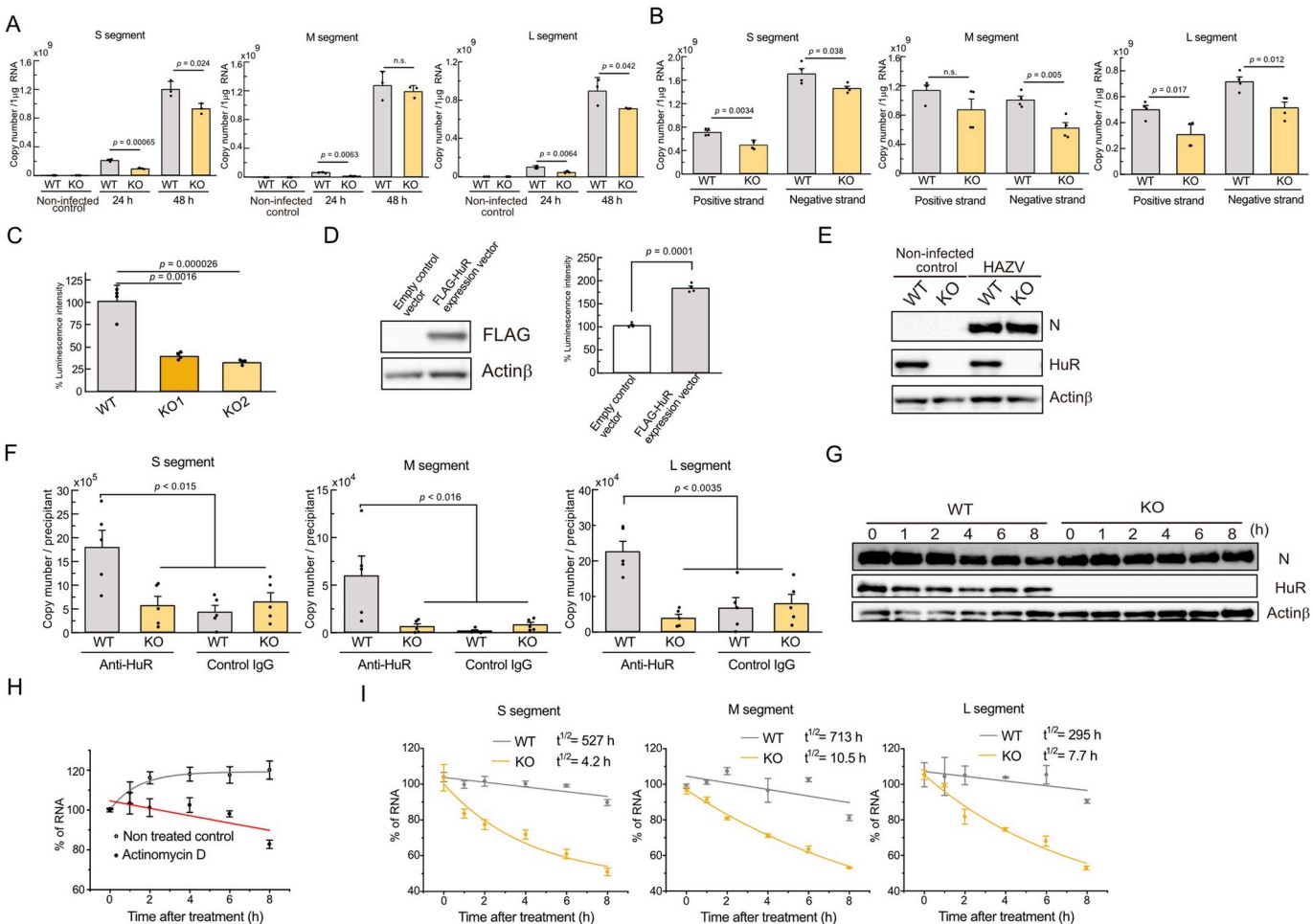

**Fig 4. HuR stabilizes HAZV RNA.** A, WT and HuR KO RAW264.7 cells were infected with 0.1 MOI HAZV, and RNA inside cells was isolated. RNA was transcribed using random primers, and the copy number of RNA inside cells was measured by real-time PCR. B, WT and HuR KO RAW264.7 cells were infected with 0.1 MOI HAZV, and RNA inside cells was isolated at 48 h post infection. RNA was transcribed using specific primers for positive or negative strands of the S, M, and L segments, and the copy number of RNA inside cells was measured by real-time PCR. C, WT and HuR KO HEK293 cells were transfected with HAZV L and N protein expression plasmids and reporter RNA, and the luciferase activity of secNluc in the supernatant was measured. D, HAZV minigenome replication in HEK293 cells was measured in the presence of control or FLAG-HuR expression vectors. FLAG-HuR expression was confirmed by western blotting with anti-FLAG antibody (left) and the luciferase activity of secNluc in the supernatant was measured (right). E, WT and HuR KO RAW264.7 cells were infected with 0.1 MOI HAZV, and cell lysates at 48 h post-infection were blotted with the indicated antibodies. F, HAZV RNA in the cell lysates of WT and HuR KO RAW264.7 cells at 48 h post infection with HAZV was precipitated with anti-HuR or control IgG antibody. RNA was transcribed using random primers and copy numbers in the precipitants were measured by real-time PCR. G, WT and HuR KO RAW264.7 cells were infected with 0.1 MOI HAZV and cells at 48 h post infection were treated with actinomycin D for the indicated time. Cell lysates were blotted with the indicated antibodies. H, RAW264.7 cells at 48 h post-infection with HAZV were treated with or without actinomycin D, and the S segment of RNA inside cells at the indicated time points was measured by real-time PCR. I, WT and HuR KO RAW264.7 cells at 48 h post-infection were treated with actinomycin D, and S, M, and L segments of RNA inside cells at the indicated time points were measured by real-time PCR. Unpaired two-tailed *t*-test (A,B,D); n.s., not significant. Ono-way ANOVA with Tukey's multiple comparison test (C, F).

of S and L segments at 48 h post infection in HuR KO cell was lowered than that in WT cells. To separate the positive and negative strands of three segments, RNA isolated at 48 h post infection was transcribed using specific primers for the positive and negative strands of S, M, and L segments, and the copy number of each segment was calculated (Fig 4B). The copy numbers of the positive and negative strands of HAZV RNA in HuR KO cells, except for the positive strand of M segment, was lower than those in WT cells. We next performed a minigenome assay to confirm viral replication. The reporter RNA consisting of the luciferase flanked by the NCR of the S segment of the HAZV JC280 strain was transfected with an

expression plasmid for HAZV L and N proteins in HEK293 cells [15]. The luciferase activity in HuR KO cells was significantly lowered than that in WT cells (Fig 4C). We tested the overexpression of HuR in a minigenome assay and found that HuR expression increased the luciferase activity (Fig 4D). HuR binds to the target host mRNA of AU-rich sequences, and increases mRNA stability. We investigated whether HuR binds to the HAZV RNA. WT and HuR KO HEK293 cells infected with HAZV for 48 h were lysed, and N protein expression was confirmed with western blotting against anti-N antibody (Fig 4E). Then, RNA in the cell lysate was precipitated with anti-HuR or control IgG antibody and the copy number of each segment in the precipitants was calculated. The RNA was precipitated by anti-HuR antibody in WT cells but not in HuR KO cells. The RNA was slightly precipitated by the control IgG antibody in both WT and HuR KO cells (Fig 4F). To investigate the RNA stability in WT and HuR KO cells, we tested the effect of actinomycin D, an inhibitor for mRNA transcription. RAW264.7 cells at 48 h post-infection with HAZV were treated with actinomycin D for the indicated time, and N protein expression in WT and HuR KO cells was constant during the treatment (Fig 4G). Actinomycin D is an anticancer drug, and cell viability was tested by the LDH release assay (S3A Fig). The LDH concentration did not increase significantly during treatment with 2.5 μg/mL actinomycin D for 8 h. Then, time-dependent changes of S segment RNA inside cells were measured by real-time PCR (Fig 4H). Untreated control cells showed an increase in viral RNA, whereas 2.5 μg/mL actinomycin D-treated cells showed a gradual decrease in viral RNA, indicating that RNA replication was restricted by actinomycin D treatment (Fig 4H). Next, we tested whether HAZV replication was inhibited by actinomycin D. RAW264.7 cells were treated with the indicated concentration of actinomycin D for 48 h, and the release of LDH was measured (S3B Fig). Treatment with 200 ng/mL actinomycin D induced LDH release, whereas treatment with < 20 ng/mL actinomycin D did not induce LDH release. RAW264.7 cells infected with HAZV were treated with the indicated concentrations of actinomycin D, and the virus titer was measured 48 h after infection (S3C Fig). HAZV titer was lowered by actinomycin D treatment. Then, WT and HuR KO RAW264.7 cells infected with HAZV at 48 h were treated with 2.5 μg/mL actinomycin D, and the $t^{1/2}$ of S, M, and L segments of viral RNA was calculated. RNA of all three segments was destabilized in HuR KO cells, compared to that in WT cells (Fig 4I).

Our findings suggest that HuR interacts with HAZV RNAs to stabilize them, and supports the replication of HAZV. We then tested whether HuR suppresses CCHFV replication. The reporter RNA consisting of luciferase flanked by the NCR of L segment CCHFV Kosova Hoti strain was transfected with an expression plasmid for CCHFV L and N proteins to WT and HuR KO HEK293 cells. The luciferase activity in HuR KO cells was significantly lower than that in WT cells (Fig 5A). The luciferase activity was increased in HuR expressing HEK293 cells (Fig 5B). HAZV and CCHFV are negative-stranded RNA viruses; therefore, positive-strand RNA is synthesized during viral replication. To clarify whether HuR binds to which strand of vRNA in the cells, HEK293 cells were transfected with *in vitro* synthesized positive- or negative-strand reporter RNA of CCHFV, and RNA in cell lysates was precipitated with anti-HuR or control IgG antibody. The reporter RNAs in the precipitants were quantified by real-time PCR. Both positive- and negative-strands bound to HuR; however, the positive-strand reporter RNA was more strongly associated with HuR than the negative-strand reporter RNA (Fig 5C). Then, positive-strand reporter RNA was transfected into WT and HuR KO HEK293 cells, and RNA in cell lysate was precipitated with anti-HuR or control IgG antibody. Positive-strand reporter RNA in WT cells was enriched in the precipitant using the anti-HuR antibody (Fig 5D). NCR in L, S, and M segments contains U-rich consecutive sequences (S4 Fig). We focused on two U-rich sequences in L segment and generated a series of deletion mutant reporter RNA lacking the 3'-NCR sequences 12029–12032 (Δregion1) and 12088–12090 (Δregion2) (Fig 5E). To clarify the

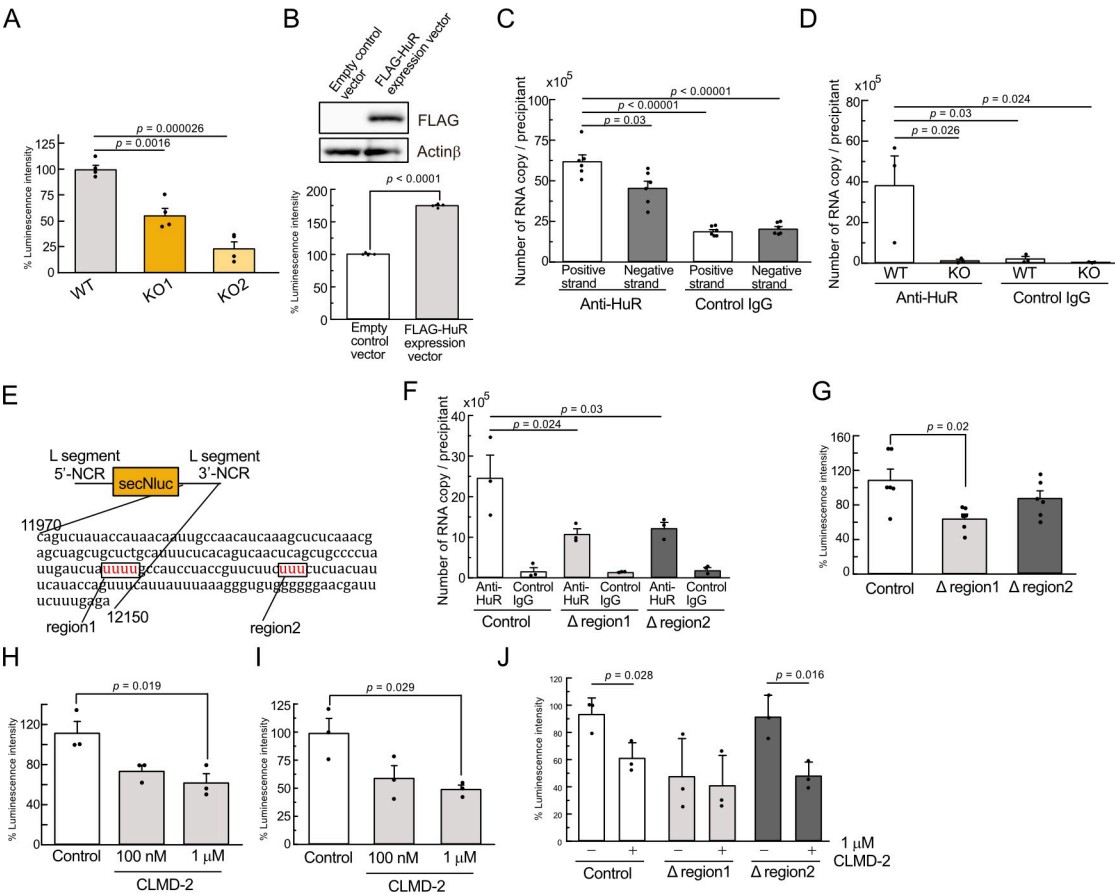

**Fig 5. HuR associates with L segment of CCHFV RNA to support minigenome replication.** A, WT and HuR KO HEK293 cells were transfected with CCHFV L and N protein expression plasmids and reporter RNA, and the luciferase activity of secNluc in the supernatant was measured. B, CCHFV minigenome replication in HEK293 cells was measured in the presence of control or FLAG-HuR expression vectors. FLAG-HuR expression was confirmed by western blotting with anti-FLAG antibody (upper panel) and the luciferase activity of secNluc in the supernatant was measured (lower panel). C, Positive and negative strands of the CCHFV reporter RNA were synthesized and transfected into HEK293 cells for 3 h. RNA in the cell lysate was precipitated using anti-HuR or control IgG antibody. The reporter RNA in the precipitants was transcribed using random primers and the copy number of reporter RNA was measured by real-time PCR. D, RNA in the cell lysate from WT and HuR KO HEK293 cells at 3 h post-transfection with the positive strand of the CCHFV reporter RNA was precipitated with anti-HuR or control IgG antibody. The reporter RNA in the precipitants was transcribed using random primers and the copy number of reporter RNA was measured by real-time PCR. E, Schematic diagram of positive-strand reporter RNA. AU-rich regions [12029–12032 (region1) and 12088–12090 (region2)] were highlighted in the 3′-NCR sequence. F, Positive strand of control, 12029–12032 (Δregion1) and 12088–12090 (Δregion2) reporter RNA were transfected into HEK293 cells, and RNA in the cell lysate was precipitated with anti-HuR or control IgG antibody. G, HEK293 cells were transfected with CCHFV L and N protein expression plasmids with reporter RNA for control, Δregion1 and Δregion2, and the luciferase activity of secNluc in the supernatant was measured. H, I, Minigenome replication of CCHFV (H) and HAZV (I) in HEK293 cells was measured after treatment with CLMD-2. J, Minigenome replication of control, Δregion1 and Δregion2 was measured after treatment with CLMD-2. Ono-way ANOVA with Tukey's multiple comparison test (A,B, C, D, F, G,H, I). Unpaired two-tailed *t*-test (B,J).

involvement of these sequences in HuR binding, control or mutant positive-strand reporter RNA was transfected into HEK293 cells and RNA in cell lysis was precipitated with anti-HuR or control IgG antibody. The control reporter RNA bound to HuR; however, both deletion reporter RNA lowered the affinity to HuR (Fig 5F). Negative-strand RNA of control and deletion mutants with L and N proteins were transfected into HEK293 cells, and the luciferase intensities were measured, and Δregion1 was significantly reduced the luciferase activity (Fig 5G). To test the inhibitory effect of HuR on CCHFV and HAZV mini-replicon replication,

HEK293 cells were treated with CLMD-2, a HuR inhibitor, after transfection and the luciferase intensities were measured. CLMD-2 treatment suppressed the CCHFV and HAZV mini-replicon replication and the HAZV titer (Figs 5H,5I,5J and S5).

## Discussion

Here, we found that HuR binds to the AU-rich region of the 3'-NCR of the L segment of CCHFV in a mini-replicon system. Previous reports have shown that HuR positively or negatively regulates viral RNA replication. HuR exhibits an antiviral effect against Zika virus [39] and assists in the assembly of the replication complex on the Hepatitis C viral- 3'UTR, and its depletion hampers viral replication [40,41]. JEV and TBEV are flavivirus, which are the same genus with Zika virus. As HuR showed an antiviral effect against Zika virus, cells were infected with JEV and TBEV compared to HAZV replication. Our results showed that HAZV replication was lowered by HuR deficiency, whereas JEV replication was increased in HuR KO cells and TBEV replication was reduced only at 72 h (Fig 1D). These results suggested that HuR tends to exhibit antiviral effects against flaviviruses. We investigated the innate immune response after viral infection, and found that *Ifnb* and *Il6* expression was increased by HAZV infection (Fig 2A), but did not increase after JEV infection (S2 Fig). As cytokine production was not robustly increased by JEV infection, innate immune activation by JEV infection is unlikely to be linked to its replication. Stress granule formation suppresses viral replication and HuR is a complex-forming protein. The antiviral effect of HuR against Zika virus was suggested to involve the formation of stress granules [39]. The antiviral effects against JEV and TBEV may also be related to HuR-mediated stress granule formation, and RNA stability by HuR does not seem to be related to viral replication.

We found that innate immune response was increased by HAZV infection and lowered by HuR deficiency (Fig 2). As innate immune activation increases antiviral effects, innate immune inhibition due to HuR deficiency is expected to enhance HAZV replication, however, the HAZV titer in HuR KO cells was lower than that in WT cells. We investigated the role of innate immune activation by HAZV infection during its replication. Innate immune response in WT MEFs after HAZV infection was increased, but not in *Rig-I/Mda5*, *IPS1* and *Tbk1/Ikk-i* KO MEFs (Fig 3A), and HAZV replication in WT MEFs was similar to that in these KO MEFs, indicating that innate immune activation by HAZV infection did not alter viral replication (Fig 3B and 3C). In contrast, pre-poly(I:C) stimulation lowered viral replication in WT MEFs, but not in those KO MEFs (Fig 3B and 3C). It is reported that ovarian tumor (OTU) domains of CCHFV and HAZV interfere with host innate immune system via the ubiquitin protease activity [42,43] and HAZV N protein interferes with the binding of TRIM25 to RIG-I and subsequent activation of RIG-I [44]. Innate immune activation by HAZV is lowered by these viral proteins. These results suggest that robust innate immune activation by pre-poly(I:C) stimulation lowers HAZV replication; however, weak innate immune activation by HAZV infection does not affect its replication.

We hypothesized that HuR associates with the HAZV genome and supports viral replication. In support of this hypothesis, HAZV genomic RNA and minigenome replication were reduced in HuR KO cells (Fig 4A and 4C), and three segments of RNA were associated with HuR (Fig 4F). The amount of cellular mRNA or viral RNA is balanced between its synthesis and degradation. The stability of host mRNA was tested after blocking transcription with a high concentration of actinomycin D (~2.5 μg/mL) [31]. The treatment of HAZV infected cells with actinomycin D also suppressed viral RNA replication in the cells (Fig 4H). HAZV replication is suppressed by the RdRp inhibitor favipiravir or ribavirin [15,45]; however, these drugs do not show acute suppression of viral RNA replication. These drugs are prodrugs that

need to be metabolized and converted into pharmacologically active drugs and take several hours to effectively suppress replication. Therefore, we compared vRNA stabilization after actinomycin D treatment and found that all three segments were destabilized in the HuR KO cells (Fig 4I). These results indicate that HuR associated with the HAZV genome to stabilize it, which aids viral replication.

Consistent with the HAZV results, the minigenome of CCHFV also showed reduced replication in HuR KO cells (Fig 5A), suggesting that HuR binds to CCHFV RNA for stabilization, which supports its replication. Further analysis demonstrated that HuR associated with the 3′-NCR of CCHFV positive-strand RNA rather than negative-strand RNA (Fig 5C and 5D). Bunyaviridae genomic RNA, including HAZV and CCHFV RNA, is partially complementary nucleotide sequences (~10 bp; the viral specific promoter element 1[PE1]) at the 5′- and 3′-NCR termini that form a double stranded RNA, panhandle structure [14,46]. The panhandle structure is critical for the circulation of viral RNA and the binding sites for N and L proteins [47]. In addition to PE1, HAZV RNA has a second complementary sequence (~20 bp; PE2) next to PE1, and CCHFV RNA also contains the PE2 sequence [14]. 3′-NCRs in S, L, and M segments of HAZV and CCHFV contain AU-rich sequences, and the AU-rich sequences of region1 and region2 in L segment of CCHFV are located at a distant location from PE1 and PE2 sites (Fig 5E). Both the positive-strand of Δregion1 and Δregion2 reporter RNAs lowered the association with HuR (Fig 5F). Minigenome replication by Δregion1 and Δregion2 reporter RNAs tended to decrease; however, only Δregion1 was significantly reduced (Fig 5G). Our results indicated that region1 in 3′-NCR contributed to HuR-mediated CCHFV replication, however AU-rich sequences may interfere with formation of the panhandle structure and association with the L protein to panhandle structure. AU-rich regions are observed in S and M segments of CCHFV 5′-NCR (S4 Fig); however, 5′-NCR and the coding region in the CCHFV genome may contribute to CCHFV replication by supporting viral RNA stabilization through HuR association. We have shown by IP experiments using cell lysates that other proteins may be involved in RNA-HuR complex formation. HuR expression was increased by the stimulation with poly(I:C) [28] and HuR expression was also increased by HAZV infection which may support HAZV replication by forming RNA-HuR complex (Fig 2D). HuR targets many host mRNAs, and HuR deficiency induces broad downregulation of cellular pathways that may affect HAZV and CCHFV minigenome replication.

In this study, we found that HuR regulates HAZV replication by the association with its RNA genome. Furthermore, the minigenome assay for CCHFV supported the hypothesis that HuR participates the replication of CCHFV. Our results suggest that HuR helps the replication of CCHFV by associating with the AU-rich region in the 3′-NCR of its genomic RNA. Treatment with the HuR inhibitor CLMD-2 reduced CCHFV minigenome replication (Fig 5H,5I and 5J). Our results suggest that RBPs are new therapeutic targets for CCHFV restriction, and the combination of RdRp and RBP inhibitors could be a new therapeutic target. However, HuR and CCHFV genome interactions were tested in the L segment of 3′-NCR RNA, and the contributions of S and M should be tested in the future. Furthermore, the role of HuR in CCHFV replication was only tested using HAZV infection and CCHFV minigenome assays; therefore, these findings need to be confirmed with CCHFV infection.

## Supporting information

**S1 Table. gene name of mutant RAW264.7 cells, related to Fig 2B.** WT and mutant RAW264.7 cells were infected with 0.1 MOI HAZV and the virus titers in the supernatants were measured at 48 h post-infection. The virus titers of the mutant cells, gene name for target

gRNA and clone number were listed.
(XLSX)

**S2 Table. Primer sequences that used in PCR and reverse transcription.**
(XLSX)

**S1 Data. Excel sheet containing raw data of figures.**
(XLSX)

**S1 Fig. Proliferation of WT, HuR KO1 and KO2 RAW264.7 cells.** $1.2 \times 10^5$ of WT, HuR KO1 and KO2 RAW264.7 cells were plated on 24 well plate and the number of cells per well was counted on the indicated day.
(TIFF)

**S2 Fig. Cytokine expression after JEV infection in wild type, HuR KO1, KO2 RAW264.7 cells.** WT and HuR KO RAW264.7 cells were infected with 10 MOI JEV, and cytokine gene expression was measured at 9 h after infection by real-time PCR. Gene expression was calculated as fold increase compared to uninfected control WT cells.
(TIFF)

**S3 Fig. The virus titer after HAZV infection during actinomycin D treatment.** A, WT and HuR KO RAW264.7 cells were treated with 2.5 μg/mL of actinomycin D and LDH release at the indicated time points was measured. B, RAW264.7 cells were treated with the indicated concentration of actinomycin D for 48 h and LDH release in the supernatant was measured. C, RAW264.7 cells were infected with 0.1 MOI HAZV in the presence of the indicated concentration of actinomycin D. The virus titer was measured at 48 h post infection.
(TIFF)

**S4 Fig. The sequence of 3'-NCR in the S and M segments of CCHFV Kosova Hoti.** The AU-rich sequences that are not located in the panhandle structures were highlighted in S and M segments of 3'-NCR.
(TIFF)

**S5 Fig. HAZV titer during CLMD-2 treatment.** RAW264.7 cells were treated with CLMD-2 after infection with 0.1 MOI HAZV. The virus titer in the supernatant at 2 days post infection was measured by the plaque assay.
(TIFF)

## Author Contributions

**Conceptualization:** Takumi Kawasaki.

**Data curation:** Takumi Kawasaki.

**Formal analysis:** Takumi Kawasaki.

**Funding acquisition:** Norisuke Kano, Minato Hirano, Kentaro Yoshii, Taro Kawai, Takumi Kawasaki.

**Investigation:** Moe Ikegawa, Mizuki Fukuta, Minato Hirano, Takumi Kawasaki.

**Methodology:** Kentaro Yoshii.

**Project administration:** Takumi Kawasaki.

**Resources:** Norisuke Kano, Daisuke Ori, Minato Hirano, Roger Hewson, Kentaro Yoshii, Taro Kawai.

**Supervision:** Takumi Kawasaki.

**Validation:** Takumi Kawasaki.

**Writing – original draft:** Moe Ikegawa, Takumi Kawasaki.

**Writing – review & editing:** Norisuke Kano, Daisuke Ori, Minato Hirano, Roger Hewson, Kentaro Yoshii, Taro Kawai, Takumi Kawasaki.

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
