## [Decision Letter · Decision Letter 0]

26 May 2024

Dear Associate Professor Kawasaki,

Thank you very much for submitting your manuscript "HuR (ELAVL1) supports viral replication by the association with non-coding region of CCHFV RNA genome" for consideration at PLOS Neglected Tropical Diseases. As with all papers reviewed by the journal, your manuscript was reviewed by members of the editorial board and by several independent reviewers. In light of the reviews (below this email), we would like to invite the resubmission of a significantly-revised version that takes into account the reviewers' comments. 

The reviewers raises several important questions that all need to be addressed.

We cannot make any decision about publication until we have seen the revised manuscript and your response to the reviewers' comments. Your revised manuscript is also likely to be sent to reviewers for further evaluation.

Sincerely,

Jonas Klingström

Academic Editor

David Safronetz

Section Editor

The reviewers raises several important questions that all need to be addressed.

Reviewer's Responses to Questions

**Key Review Criteria Required for Acceptance?**

**Methods**

-Are the objectives of the study clearly articulated with a clear testable hypothesis stated?

-Is the study design appropriate to address the stated objectives?

-Is the population clearly described and appropriate for the hypothesis being tested?

-Is the sample size sufficient to ensure adequate power to address the hypothesis being tested?

-Were correct statistical analysis used to support conclusions?

-Are there concerns about ethical or regulatory requirements being met?

Reviewer #1: Objectives of the study are not clearly articulated and the aim of the work is not well indicated in the introduction. The picture of the study became clear only after the reading of the discussion.

Authors used several cellular and virus models but a better connection between the experiments should be reported. In addition, three different viruses were compared in the first part of the work showing a different behavior in their replication HuR-mutated cells. Thus, to support the specific mechanism of action of HuR on HAZV replication, in some experiments all the three viruses should be included, as reported in the general comments.

Some technical details are lacking, as indicated in the general comments.

The statistical analyses applied to the results is sufficient.

Reviewer #2: 1. The study objectives are clearly stated

2. The exact experimental details (cell line, MOI, time point post infection/transfection/stimulation) are sometimes somewhat difficult to find. Recommend putting all these details in both the Figure legends and the Methods section, and in the main text where appropriate. 

3. In Figure 4 authors assess viral RNA stability using Actinomycin D. Actinomycin D blocks cellular transcription, but I am not certain if it affects replication of negative strand RNA viruses like HAZV. It certainly does not affect replication of +RNA virsues as ActD is routinely used to label viral RNA in the absence of host transcription. Since ActD is well tolerated by most cell lines, authors should add some later time points and viral titers to more convincingly show that ActD inhibits viral replication. The presented data using ActD shows some inconsistencies, as in Fig 4D the HAZV RNA levels remain steady during 4h ActD treatment, whereas in Fig 4E HAZV RNA levels decreased in ActD treated infected WT cells.

Reviewer #3: line 206-217: refer somewhere to S1 table

Fig 3c: Add in the method the qty of poly(I-C) used, how long was the pre and post-treatment, when did you recover the supernatant? how did you purify the supernatant from poly(I-C) before titration to avoid the poly(I:C) to activate the cells you were using for your titration? 

several figures: Explain in the method how you calculated the relative RNA binding: do you have a standard RNA for your RT-PCR to determine the number of copies? Or?

Please note the n for each figure in the legend. By eyes, it looks like that the number of repeat in the same experiment is not the same sometimes. For example, Fig 5B, anti HuR, it looks like n=6 for the positive strand while it seems to be 4 in the negative strand. Fig 5G, it seems that n=4 for control but n=3 for the treated ones. If it is the case, please repeat the experiments to have the same number of repeat for each condition.

**Results**

-Does the analysis presented match the analysis plan?

-Are the results clearly and completely presented?

-Are the figures (Tables, Images) of sufficient quality for clarity?

Reviewer #1: The analyses plan was poorly described, thus the match of the presented analyses with the analyses plan cannot be evaluated. Not all the results are clearly presented and some controls are lacking as well as additional analyses to support authors’ conclusions increasing the strength of the study. Something is lacking to have a complete picture of the phenotype observed in the mutated cell lines linking the modulation of the immune response to HAZV replication.

The quality of images is sufficient for clarity.

Reviewer #2: 1. While I don’t doubt that HuR plays a role in the HAZV replicative cycle, the data demonstrating that HuR is supporting HAZV/CCHFV replication by directly binding and stabilizing viral RNA is not very strong, especially the mechanistical side of it. HuR KO only has a modest effect on viral titers (0.5-1.0 log reduction), and knock-in experiments demonstrating that ectopic expression of HuR relieves the negative effect of its depletion have not been done. A lot of the presented data is somewhat circumstantial. E.g. authors state that HuR bind directly to HAZV RNA, but since these are IP experiments using cell lysates, it could be a protein complex binding to the RNA. 

2. Including the section in the introduction addressing innate immunity (line 81-90) and Fig 2+3 does not fit well with the HuR/HAZV story, resulting in a disjointed manuscript. The premise presented in Fig 2, describing that depletion of HuR results in dampened immune responses was already published by the authors, so this is not novel and not specifically related to HAZV infection. Additionally, this does not explain the mechanism(s) by with HuR supports HAZV replication, as the expectation is that lower immune responses would result in higher HAZV titers. 

3. Fig 3: while interesting in itself, this figure does not contribute to the HuR story and does not fit in well with the data presented in Fig 1, 4 and 5. 

4. Demonstrating the interaction of HuR with HAZV/CCHFV RNA is a pivotal figure in this manuscript (Fig 4C, Fig 5B/E). Authors pull down viral RNA from WT and HuR KO cells using a HuR antibody. The presented difference looks impressive but obviously the absence of HuR in the KO cells will preclude any RNA immunoprecipitation. Additionally, authors already demonstrated that viral RNA levels are lower in HuR KO cells. Authors should include antibodies against NP as a positive control to demonstrate that HAZV/CCHFV RNA is indeed present in somewhat similar levels before IP. 

5. Does CLMD-2 inhibit HAZV infection?

Reviewer #3: Figure 1: what about HAZV RNA level inside the cells?

line 224: please explain why you made a kinetic of hazv infection on raw264.7 and MEF

line 227: viral titer doesn’t decrease at 72h for raw264.7

line 229: confusing. move and rephrase the part “and virus titer...plaque assay” after “...S1 table.” and move “fig 1B” after “highlighted” line 233 

line 232: is it “both” cell lines or “the” cell lines? it seems it is “the” cell lines.

line 234: why did you choose only these hits?? it is not because they were the highest or the lowest compare to wild-type as for example, they are 2 cells lines between zfp and Parp.

line 235: please precise if JEV and TBEV are control and explain why you choose them as control

line 236: add “48hpi” after “were measured”

Line 238: you used JEV and TBEV and show the data but nothing is discussed. Please discuss the data, or you remove the data.

Line 242: what about the other hits? why did you choose only HuR? Explain

Line 245-248/fig 2A: Why did you infect with MOI 1 for 9h while until now you used MOI 0.1 and 48hpi? Even in your kinetic on wt and ko cells you do not have 9h time-point. 

Line 248-250: you changed again the MOI to be back to MOI 0.1 and here you go for day 1, 2 and 3pi. Nothing can be compared as MOI 1 9h, there is a decrease in IL-6 in KO cells while at MOI 0.1 the significant decrease is 3dpi. Both data together seems not to match.

Line 250-253: Back to MOI 1. 

line 253: IRF3 phosphorylation level begun to increase from 9h in wild-type cells but was less increased in HuR KO cells. Then you show data for IRF3, pp65 and p65 but nothing in the text. please write a sentence to give results for them.

line 255: Why changing the cells to hek293? please explain. Then there is no data on the kinetic of the virus on these cells. Which MOI and why 24pi?

line 257: not suppressed but reduced.

There is a lack of logic in your data: you show data for several cytokines, then you focus on Il-6 (why particularly Il-6?) then you go to IRF3/p65 to come back to IFN-β. What about TNF, Cxcl10 or Ccl2??

line 259: HAZV infection triggers activation of innate immune response in mouse macrophages.

line 260: “...to be recognized by RIG-I/MDA5”. Add “and HuR regulates innate immune response via RIG-I/MDA5-dependent... pathways” that is already noted line 242 to make it more understandable.

line 261/fig 3A: why do you change again the cell line? note on the figure that – and + mean – infection or + infection. Which MOI? note in the legend the MOI and the time-post-infection. Why 24h while the pick of viral titer in MEK is 48hpi? Explain.

line 264: significantly increased in WT infected cells contrary to KO cells

Figure 3B: back to MOI 0.1 48hpi. what about the RNA inside the cells?

line 268: indicating the innate immune response doesn’t affect virus production.

line 269/fig 3c: back to raw264.7. why? Also explain the role of poly-(I:C). Add in the method the qty of poly(I-C) used, how long was the pre and post-treatment, when did you recover the supernatant? how did you purify the supernatant from poly(I-C) before titration to avoid the poly(I:C) to activate the cells you were using for your titration? Please note statistical analysis comparing pre and post poly(I-C) treatment. It seems the data for HuR KO cells are not significant, you cannot say that “pre-stimulation with poly(I-C) in both WT and HuR KO cells reduced the virus tirus compared to the control or post-poly(I-C) (line 272). Please rectify. 

line 270: is there any paper showing that the protocol you used for innate immune response stimulation really stimulate the immune response. if yes, please cite. if not, please measure the cytokines response under poly(I-C) treatment.

line 280/fig 4B: you changed again the cells to Hek293. Please justify. Change “to test” by “to confirm”.

line 289: “was not precipitated” to change by “slightly” or similar. Explain in the method how you calculated the relative RNA binding: do you have a standard RNA for your RT-PCR to determine the number of copies? Or?

Line 295: control cells showed an increase in viral RNA amount whereas ... suppressed viral RNA replication” or “de novo synthesis”.

Line 316: “NCR in L segment...”. But you used the S segment for HAZV previously. If it is for all NCR please change “L segment” to “L, S and M segments” or similar.

Line 327: on CCHFV mini-replicon replication and not on CCHFV replication. Same line 330

**Conclusions**

-Are the conclusions supported by the data presented?

-Are the limitations of analysis clearly described?

-Do the authors discuss how these data can be helpful to advance our understanding of the topic under study?

-Is public health relevance addressed?

Reviewer #1: Conclusion are partially supported by the results and not limitations are described.

The advance of the knowledge and the potential applications are reported, as well as the potential application for the development of therapeutics.

Reviewer #2: 1. Since the majority of the experiments are done with Hazara virus, the title/abstract/author summary/introduction should reflect this. Only the last figure uses CCHFV, and this is restricted to the use of a minigenome assay (i.e. no infectious CCHFV was tested). 

2. The discussion section is largely repeating the results section, rather than placing their results in a larger context. HuR has been implicated in the replicative cycles for many viruses (including JEV). This should be more extensively addressed in the Discussion. 

3. Authors do not address limitations of their study in the Discussion.

Reviewer #3: The discussion has to be modified: in general, the discussion needs to go a bit deeplier and try to give more explanation then just repeating the results. More of that, the conclusions go to far and have to be tempered.

Line 333: we found that HuR binds to the AU-rich region of the 3’-NCR of the L segment of CCHFV in a mini-replicon system.

line 337: not suppress but reduce

line 338 “it increasesd JEV replication” And? any explanation?

line 342: not suppressed, but “...induce expression of some cytokine gene, which were strongly reduced by HuR deficiency”

line 345: “comparable to wt cells” instead of “comparably increased.”

line 348-350: it is results but no explanation. please give some possible explanation.

line 381: significantly reduced, not suppressed.

line 388: it is not required, it participates.

line 390: CLMD-2 doesn’t suppress CCHFV minigenome replication but reduces (50% decrease is far from suppression)

line 391: could be new therapeutic target. 

line 392: finish the discussion by explaining that all the findings have to be validated in a real infection study using CCHFV.

**Editorial and Data Presentation Modifications?**

Reviewer #1: Minor comments:

In the section “Introduction” avoid the repetition of the segmented nature of the genome. Write better the fact that CCHFV and HAZV belong to the same genus and share many structural and biological characteristics. 

Line 76: the M segment encodes the precursor that originates the mature glycoproteins of the viral envelope (Gn and Gc) and some additional proteins.

Lines 81-83: it is not clearly exposed the immune-regulation and its link to the next section

Line 102: clarify the mean of “sequence method”

Line 126: the reference 21 is not appropriate for this citation

Line 133: the word “mounted” is not appropriate

Lines 146-158: change “sense” with “forward”

Line 178, the word “harvested” is not appropriate

Lines 235-236: Sentence is wrong. Cells were infected with the viruses. Please rewrite.

Reviewer #2: 1. Overall, this manuscript would benefit from substantial rewriting. The English language is in places insufficient. Various sentences are unclear or confusing. 

2. Please order the methods section in order of appearance in the manuscript

3. Line 133: Please provide more information on the TBEV/JEV strains used

4. Line 139-159: please separate this into separate sections describing RNA isolation, the RNA stability assay and RT-PCR assay.

5. Line 146-159: please provide the primer sequences as a supplementary table

6. Line 162 (ELISA methods section): please specify here what cells were used.

7. Line 180: please indicate used MOI and how long after infection luciferase activity was measured.

8. Line 224-226: The results section starts very abrupt. It would be helpful if it started with a statement of what/why this was done. It is unclear why MEF cells were infected alongside RAWs.

9. Figure 1: The absence of HuR expression was not confirmed for the RAW264.7 cells. 

10. Line 239-241: please rewrite to enhance clarity 

11. Line 242-245: It is unclear how HAZV/CCHFV OTU domains are linked to HuR and why authors looked at the induction of various immune genes. 

12. Line 246: please specify that RNA levels are measured

13. Fig 2A should also assess HAZV RNA levels.

14. Fig 2A: It is unclear what the y-axis reflects. Is this fold increase compared to uninfected cells? Same question for Fig 2E.

15. Fig 2C: HuR protein levels seem to be upregulated in infected cells? This would be interesting to pursue more. 

16. Fig 3A: it is unclear how long after HAZV infection gene expression levels were determined, or what MOI was used. 

17. Fig 4: Relative viral RNA levels are relative to what? Control is uninfected?

Reviewer #3: Abstract

Rephrase abstract lines 33-36 as it is not clear from cchfv to hazv even if you say that hazv used as a model before and is that the screening that was done using a mutant cells lines? please make it clearer (the author summary is clearer)

Author summary

Line 56: In the author summary: hur inhibitor doesn’t suppress cchfv minigenome replication. 50% is not suppressing, it is reducing

Introduction

Line 111: HuR is not required for the replication of CCHFV: you still have 25 to 50% luciferase activity in KO cells (fig 5A)

**Summary and General Comments**

Reviewer #1: The manuscript by Ikegawa et al reports the identification of HuR as a host protein required for CCHFV and HAZV replication in cells by its ability to bind and stabilize viral RNA.

The topic is of interest providing a new piece of knowledge on the virus-host interaction. However, there are important issues that need to be addressed.

The abstract and the summary contain non-homogeneous information and should be rewrite. In addition, the abstract start with HAZV and its role as model for CCHFV. In the introduction no info on HAZV were reported. I suggest to add a paragraph on HAZV. In addition, more details on the CCHF pathogenesis should be introduce to better understand the correlation between the different parts of the introduction (immune activation, target cells of CCHFV, etc). 

In the section “material and methods” some details are missing, such as: the indication of JEV and TBEV strains, and their production; details on the PCRs protocol; the antibodies used in the western blotting; the protocol of poly(I:C) stimulation.

Regarding the “results” section:

Since viruses are strongly affected by cell metabolism and growth, data on the proliferation of mutated and wild type cells should be reported.

In Tab S1 there are three genes that, after inactivation, are associated to a higher level of inhibition of HAZV in comparison to HuR. Why was HuR selected instead of the other three genes? 

Lines 242-245. Regarding the mechanisms of HAZV immune-evasion, the N protein is also involved (10.3390/v14091965). This information should be included and discussed. 

Figure 1D. Since TBEV and JEV are differentially affected by the inactivation of HuR, to better support the specific down modulation of HAZV replication, the kinetics of TBEV and JEV should be included as comparison.

Results in figure 2A lack of a negative control represented by the inactivated virus to evaluate the effect of the productive infection (with virus replication) in comparison to the input of an inactive virus.

Lines 257-258. The sentence stands that “These results indicated that the innate immune response against HAZV infection was reduced by HuR-deficiency.” In this scenario, the deletion of HuR should increase the replication of HAZV. The reduction of HAZV replication and immune-activation in HuR KO cells can indicate a broad down regulation of many cellular pathways. Or, the inhibition of HAZV replication, due to the HuR KO, could stimulate partially the immunity. The comparison with JEV could help to clarify the results.

Line 262. There is not information about the generation and characterization of the mutated MEF cell lines.

Why did the Authors use different cell lines for the experiments related to figures 3B and 3C? The poly I:C treatment should be included as additional control in figure 3B.

Fig4A: considering the type of mechanism under investigation, more details could be obtained evaluating the amount of positive and negative strand segments of the viral genome using specific primers for retro-transcription and/or the evaluation of viral mRNA production to discriminate between replication and transcription.

Lines 281-284. An additional control should be included, such as the western blot for the N protein of the virus. This could be useful also for data in figure 4D-4E. Viral protein is more stable than RNA and can work as internal control. 

Lines 316-318. Authors report the presence of two AU- U-rich sequences in the segment L. What is the situation of the segments S and M? Are there the same or similar sequences? If not, can the potential effect of the deletion of HuR modify the amount of S segment or viral RNA polymerases expression?

Are the mutations performed in both the genome extremities allowing the formation of the panhandle structure? 

Lines 327-330. Experiments with the inhibitor of HuR should include HAZV replicon and the mutated sequence (delta 1 and delta 2) thus allowing a better comparison.

Reviewer #2: Authors investigated the role of RNA-binding proteins during Hazara virus infection by generating 66 KO cell lines using CRISPR. HuR was identified as one of the strongest hits, of which depletion resulting in a modest reduction in HAZV titers. As the authors described in a previous publication, depletion of HuR results in an impaired activation of immune responses. Furthermore, they show that both HAZV and CCHFV minigenome activity is affected in HuR KO cells, likely due to the lack of HuR-mediated viral RNA stabilization.

Reviewer #3: This manuscript is of interest for researchers in CCHFV and give interesting preliminary data for a possible way to explore in antiviral development. Nethertheless, some modifications are required.

The titer has to be modified as you do not show that HuR supports (real) CCHFV viral replication, and please precise that is in a mini-replicon system. You can also add HAZV in the title as it is 80% of your data... 

Please put p value on top of each batch of sample and not on different batches all together. For example Fig 2A, put the significance for KO1 and KO2 separately.

Why using Raw264.7 and not human cell lines like THP-1? It would have given more relevant data, particularly for the immune response part.

For all data involving the virus, data on viral RNA level inside the cells are necessary, not only the production of viral particles, to confirm the effect of the KO of HuR in HAZV RNA replication.

PLOS authors have the option to publish the peer review history of their article (what does this mean?). If published, this will include your full peer review and any attached files.

Reviewer #1: No

Reviewer #2: No

Reviewer #3: Yes: Vanessa M. Monteil
---

## [Decision Letter · Decision Letter 1]

12 Sep 2024

Dear Associate Professor Kawasaki,

Thank you very much for submitting your manuscript "HuR (ELAVL1) regulates the CCHFV minigenome and HAZV replication by associating with viral genomic RNA" for consideration at PLOS Neglected Tropical Diseases. As with all papers reviewed by the journal, your manuscript was reviewed by members of the editorial board and by several independent reviewers. The reviewers appreciated the attention to an important topic. Based on the reviews, we are likely to accept this manuscript for publication, providing that you modify the manuscript according to the review recommendations. 

Sincerely,

Jonas Klingström

Academic Editor

David Safronetz

Section Editor

Reviewer's Responses to Questions

**Key Review Criteria Required for Acceptance?**

**Methods**

-Are the objectives of the study clearly articulated with a clear testable hypothesis stated?

-Is the study design appropriate to address the stated objectives?

-Is the population clearly described and appropriate for the hypothesis being tested?

-Is the sample size sufficient to ensure adequate power to address the hypothesis being tested?

-Were correct statistical analysis used to support conclusions?

-Are there concerns about ethical or regulatory requirements being met?

Reviewer #1: The methods section of the revised manuscript has improved since the first submission. It is acceptable.

Reviewer #3: The method part has been completed to give more details.

**Results**

-Does the analysis presented match the analysis plan?

-Are the results clearly and completely presented?

-Are the figures (Tables, Images) of sufficient quality for clarity?

Reviewer #1: Results are clearly presented and match the experimental plan. All the new experiments help to support Authors' hypothesis.

Reviewer #3: The results part has been deeply corrected/completed to make the data stronger.

**Conclusions**

-Are the conclusions supported by the data presented?

-Are the limitations of analysis clearly described?

-Do the authors discuss how these data can be helpful to advance our understanding of the topic under study?

-Is public health relevance addressed?

Reviewer #1: Conclusions are supported by results and clearly presented. The content of the conclusions meets editorial requirements.

Reviewer #3: The conclusions are now better supported by the data and are now reflecting the real scope of the data.

**Editorial and Data Presentation Modifications?**

Reviewer #1: (No Response)

Reviewer #3: (No Response)

**Summary and General Comments**

Reviewer #1: The new version of he manuscript is significantly improved and data can support the Authors' conclusions.

I just suggest to check the text for few type errors and some modifications:

line 36: I do not understand why Authors wrote (or the stability of mRNA). This is the same concept reported in the sentence

Fig. 4b: change "psitive" with "positive".

lines 366-367: I am not sure that the sentence “The reporter RNA consisting of the luciferase gene with the S segment of the NCR of the HAZV JC280 strain was transfected...” is correct. Probably it is clearer as “The reporter RNA consisting of the luciferase flanked by the NCR of the S segment of the HAZV JC280 strain was transfected...” There are other few sentences with the same form to check.

line 452: change "dose" with "does"

Reviewer #3: I would like to acknowledge the authors for taking into account all my comments and having completed/modified the figures/text accordingly. The data are now completed and the conclusions are relevant. The new version of the MS is, in my opinion, ready for publication.

PLOS authors have the option to publish the peer review history of their article (what does this mean?). If published, this will include your full peer review and any attached files.

Reviewer #1: No

Reviewer #3: Yes: Vanessa M. Monteil

Figure Files:

Data Requirements:

Reproducibility:

References

---

## [Editor Report · Decision Letter 2]

20 Sep 2024

Dear Associate Professor Kawasaki,

We are pleased to inform you that your manuscript 'HuR (ELAVL1) regulates the CCHFV minigenome and HAZV replication by associating with viral genomic RNA' has been provisionally accepted for publication in PLOS Neglected Tropical Diseases.

Best regards,

Jonas Klingström

Academic Editor

David Safronetz

Section Editor

---

## [Editor Report · Acceptance letter]

25 Sep 2024

Dear Associate Professor Kawasaki,

We are delighted to inform you that your manuscript, "HuR (ELAVL1) regulates the CCHFV minigenome and HAZV replication by associating with viral genomic RNA," has been formally accepted for publication in PLOS Neglected Tropical Diseases.

Best regards,

Shaden Kamhawi

co-Editor-in-Chief

Paul Brindley

co-Editor-in-Chief
